# The Use of 3D Polylactic Acid Scaffolds with Hydroxyapatite/Alginate Composite Injection and Mesenchymal Stem Cells as Laminoplasty Spacers in Rabbits

**DOI:** 10.3390/polym14163292

**Published:** 2022-08-12

**Authors:** Ahmad Jabir Rahyussalim, Dina Aprilya, Raden Handidwiono, Yudan Whulanza, Ghiska Ramahdita, Tri Kurniawati

**Affiliations:** 1Department of Orthopaedic & Traumatology, Cipto Mangunkusumo National General Hospital and Faculty of Medicine, Universitas Indonesia, Jakarta 10430, Indonesia; 2Stem Cell Medical Technology Integrated Service Unit, Cipto Mangunkusumo General Hospital, Jakarta 10430, Indonesia; 3Stem Cells and Tissue Engineering Research Cluster, Indonesian Medical Education and Research Institute (IMERI), Faculty of Medicine, Universitas Indonesia, Jakarta 10430, Indonesia; 4Department of Mechanical Engineering, Faculty of Engineering, Universitas Indonesia, Depok 16424, Indonesia; 5Research Center for Biomedical Engineering, Faculty of Engineering, Universitas Indonesia, Depok 16424, Indonesia; 6Mechanical Engineering and Materials Science, McKelvey School of Engineering, Washington University in St. Louis, St. Louis, MO 63130, USA

**Keywords:** scaffold, polylactic acid, hydroxyapatite, alginate, mesenchymal stem cells, laminoplasty, spacer

## Abstract

Several types of laminoplasty spacer have been used to fill bone gaps and maintain a widened canal. A 3D scaffold can be used as an alternative spacer to minimize the risk observed in allografts or autografts. This study aims to evaluate the in vivo biocompatibility and tissue–scaffold integration of a polylactic acid (PLA) scaffold with the addition of alginate/hydroxyapatite (HA) and mesenchymal stem cell (MSc) injections. This is an experimental study with a pretest and post-test control group design. A total of 15 laminoplasty rabbit models were divided into five groups with variations in the autograft, PLA, HA/alginate, and MSc scaffold. In general, there were no signs of inflammation in most samples (47%), and there were no samples with areas of necrosis. There were no significant differences in the histopathological results and microstructural assessment between the five groups. This demonstrates that the synthetic scaffolds that we used had a similar tissue reaction and tissue integration profile as the autograft (*p* > 0.05). We recommend further translational studies in humans so that this biocompatible fabricated scaffold can be used to fill bone defects.

## 1. Introduction

Cervical myelopathy is a chronic degenerative disease that results in the compression of the spinal cord at the cervical level. Various decompression methods have been developed using both anterior and posterior approaches. Laminoplasty is a lamina reconstruction technique with a posterior approach followed by lifting a portion of the lamina to make a door or hinge. Several techniques have been proposed to fill the gaps that are intentionally made to release pressure while protecting the spinal cord by using spacers from autograft spinous processes fixed with wire, an autograft from the iliac crest or costae, or biomaterials such as ceramics or hydroxyapatite [1,2,3,4,5].

Bone autograft currently served as the gold standard for treatment strategies in dealing with bone defect. It has all the properties of the ideal bone substitutes to induce bone healing a growth: osteogenic cells, osteoconductive, osteoinductive and biomechanically stable. However, the autograft also had disadvantages, such as additional surgical procedure, risk of infection, hematoma and chronic pain. The allograft had similar advantages like the autograft and does not need an additional surgical procedure, but it carries other disadvantages such as disease transmission or immune reaction from donor to recipient. Biomaterials and tissue engineering provide an alternate solution. It can be as reliable as an autograft, but it must fulfill several criteria: biophysical stability, geometrical form, surface structures, porosity and pore structures, and resorbability of the materials [6,7]. Not only that, to fill bone defects, a diamond concept of bone tissue regeneration should be applied. This concept, introduced by Giannoudis et al. [8,9,10], consists of osteoinductive substances, osteogenic cells, scaffold, and a mechanical environment that together encourage optimum bone regeneration to close the defect, which are the main focuses in the field of tissue engineering [11,12,13,14,15].

The use of scaffolds or three-dimensional containers as biomaterials provides a supporting structure for cell proliferation and differentiation. The reason for the biomaterials’ usage is to reduce the risk of tissue rejection or disease transmission, such as in the case of allografts, and to reduce the morbidity caused by autografts. Several types of scaffold material have been widely studied, including hydroxyapatite (HA), polymers, ceramics, polymer and ceramic composites, and natural polymers such as collagen and chitin [1,2,3].

Several criteria must be met before a scaffold can be implanted safely and effectively into the human body. Scaffolds are designed to be biocompatible, biodegradable, reproducible and have high porosity with interconnected gaps and mechanical properties that are similar to those of the host tissue and have no potential to cause serious immunological and rejection reactions. In addition, a scaffold is expected to mediate cell proliferation and differentiation and extracellular matrix secretion, and to carry biomolecular signals for cell communication [1].

In vitro biocompatibility had been assessed in a preliminary study by Rahyussalim et al. by a direct contact test with mesenchymal stem cells and indirect contact test scaffold with an MTT test [16]. It was proven that polylactic acid (PLA) and alginate-based scaffolds have low levels of inhibition and toxicity. Several studies have explored various types of scaffold material, but there are not many comparing several scaffold materials used as spacers in laminoplasty procedures [17,18,19,20]. Therefore, this study aims to continue the in vivo biocompatibility testing of PLA scaffolds and find out what difference an additional alginate/HA injection makes to prove in vitro biocompatibility and compare the effectiveness of using various types of spacers for laminoplasty. This study differs from the previous study [5] in that we experimented not only with the mechanical properties of the scaffold but also with its biologic effect towards living animals.

## 2. Materials and Methods

### 2.1. Animal Model

This was an experimental study with one group, and a pre- and post-test design. We used rabbits as the animal model because they are phylogenetically similar to humans [21]. The animals’ size, easy maintenance, and ease of blood collection make them suitable as research subjects. Rabbits also have a lifespan of around five to eight years, which is suitable for long-term research [22]. Several similarities have been reported in terms of the mechanical properties of rabbits’ vertebrae. They also have a quick bone remodeling ability [21,22]. The spine of a rabbit also plays an important role as a body support for posture. A rabbit’s spinal curvature bears the stress generated by body movements such as jumps or gravity [21,23].

Subjects were 3500–4000-g rabbits of the New Zealand White Specific Pathogen-Free (NZW SPF) *Oryctalagus cuniculus* strain that were acclimatized in a laboratory with a controlled temperature of 18–21 °C, 55% air humidity, a diet of commercial pellets, fresh food twice a day (5 g/100 g body weight) and water ad libitum before and during the study. The sample size was determined using the resource equation method, which is widely used for complex biological experiments with multiple variables and interventions [24,25,26,27]. Twenty New Zealand white rabbits were divided into five groups randomly (three rabbits per group), with five extra rabbits to replace dead subjects during the study. The sample was calculated using the resource equation formula [28].

The intervention of this study was the use of different materials as laminoplasty spacers. The different spacers used were autografts of the spinous process, PLA-based laminoplasty spacers, PLA-based laminoplasty spacers augmented with HA/alginate, PLA-based laminoplasty spacers augmented with mesenchymal stem cells, and PLA-based laminoplasty spacers augmented with HA/alginate and mesenchymal stem cells. This study was conducted with the approval of the University of Indonesia Faculty of Medicine’s ethical committee.

### 2.2. PLA Scaffold Fabrication

The scaffolds used in this study were six-layer poly-L-lactic acid (PLA) polymer-based scaffolds, as shown in Figure 1. The manufacturing process used modern techniques with rapid prototyping and fuse deposition modeling (FDM) methods, with modifications to the polymer material printing nozzle based on computer-assisted design (CAD) (Autodesk Inventor, Autodesk, Inc., San Rafael, CA, USA, 2013) software and computer-aided manufacturing (CAM) software (Pronterface—Slic3r, RepRap, England). The nozzle was heated to melt the material and make the scaffold thermoplastic so that it could be manipulated. The manipulation was directly controlled by CAM. The nozzle diameter was 0.5 mm and that of the PLA filament was around 1.75 mm. The extrusion parameter was set to 6 mg/s. After it was extruded in its thermoplastic form and the layers of the scaffold formed, the material hardened immediately when it cooled down.

Scaffolds were specifically designed for the porosity and modulus of elasticity to be biomimetic. The porosity was adjusted so that alginate could be added to the matrix of the PLA scaffold. The scaffolds used in this study had six layers, with a Young’s modulus of 84.42 MPa.

Preliminary studies were carried out to test the mechanical and biocompatibility properties in vitro. Biomechanical tests were performed by providing mechanical loading on certain strains, then comparing it with the bone strength (stress–strain curves). The compression modulus was tested with a Gotech AI-700 Ultimate Testing Machine (GOTECH TESTING MACHINES INC., Taichung City, Taiwan). The deformation velocity was set to 0.2 mm/min and 50% of the initial height, whereas in vitro biocompatibility was assessed by a direct contact test with mesenchymal stem cells and an indirect contact test scaffold with an MTT test (3-(4,5-dimethylthiazol-2-yl)-2,5 diphenyltetrazolium bromide). The 3D PLA scaffolds were seeded using mesenchymal stem cells (MSC). Later, the cultured cells were tested with Vybrant^®^ and the absorbance value was measured with ELISA at a 570 nm wavelength. Cell proliferation and cell adhesion to scaffolds were evaluated on the second and sixth days.

### 2.3. Hydroxyapatite (HA)/Alginate Composite Injection

We injected the hydroxyapatite/alginate at a consistent flow rate of 20 mL/h using a syringe pump. Materials used as composites were hydroxyapatite, sodium alginate, and sodium chloride. HA/alginate was prepared from brown algae with 0.9% sodium chloride (NaCl) and calcium chloride (CaCl_2_) 0.2 M (Merck, Kenilworth, NJ, USA). Sodium alginate (30% g/vol) and hydroxyapatite powder (70% g/vol) were mixed into a 0.9% NaCl solution. The solution was stirred with a magnetic stirrer for 5 min at 40 °C, until the solution formed a paste. The paste was then inserted into the syringe and injected into the PLA scaffold matrix. After the injection, the porosity of the injected scaffolds was measured using the same method by using ImageJ (Version 1.47 for Windows, 64-bit, free software, National Institutes of Health, Bethesda, MD, USA). After thresholding and binarization was performed, the porosity was determined using the image volume method. The porosity of the PLA scaffolds was 62.3% before the injection and around 48.7% after the injection.

### 2.4. Rabbit Mesenchymal Stem Cell Preparation

Cells were planted in a flask 25 (T25) surface container at a cell density of 5000 cells/cm^2^. The medium was changed and observed once every 2–3 days. Confluent cells (80%) were sub cultured until they reached the desired passage (P3). In this experiment, alpha MEM and Dulbecco’s modified Eagle’s medium (DMEM, [GIBCO 31600-034]) were used as the medium. Cells were implanted in flask 25 after they reached the desired passage (P3) and 80% confluence was reached. The cell medium was then discarded, and the cells were rinsed twice with PBS solution and placed in a flask with enzymes for incubation in a CO_2_ incubator for 5–10 min. After the cells were released, the enzymes were neutralized in 2 mL of complete culture medium. Cell suspensions were collected in centrifuge tubes and rotated at 1200 rpm for 10 min. Cell pellets were dissolved with 1 mL complete culture medium and cells were then counted.

### 2.5. Cell Seeding Preparation

Before the co-culture, scaffolds were immersed in 400 mL of a 70% alcohol solution for 20 min, in a 1.5-mL tube. Scaffolds were then rinsed with sterile double-distilled water and filtered with a 0.22-µM filter 2–3 times. After rinsing, scaffolds were ready to be placed in a culture container filled with cells.

### 2.6. Mesenchymal Stem Cell Co-Culture with Scaffold

The calculated cell suspension was then placed in a 24-well culture plate at a density of 5000 cells/cm^2^, and the cells were put into an incubator (37 °C, 5% CO_2_) for 30–60 min. After the cells attached to the surface of the culture container, scaffolds were placed in the center of the well and submerged in the culture medium. The culture container containing the cells and scaffold were put into an incubator at 37 °C, 5% CO_2_. On the second day after co-culture, the scaffold was reversed. Scaffolds were ready to be transplanted into experimental animals after the cells had been incubated for 3–4 days or had reached 80% confluence.

### 2.7. Laminoplasty Measures

The laminoplasty procedure with the open door laminoplasty (ODL) technique was performed at the level of the lumbar vertebrae to obtain a more optimal operative field in rabbits. The rabbit’s back was shaved at the lumbar level, then, rabbits were fasted for 12 h and given butorphanol tartrate (0.1 mg/kg). Before the implantation and exsanguination procedure, a blood sample was taken to determine the markers of inflammation. Rabbits were prepped and draped in a pronated position. General anesthesia was administered intramuscularly as a mixture of ketamine HCl (15 mg/kg), xylazine (2.5 mg/kg) and acepromazine maleate (0.75 mg/kg). Previously, rabbits were given atropine sulfate 1 mg/kg subcutaneously, and a small dose of Nembutal (10 mg/kg) intravenously, to provide a deep anesthetic effect. Enrofloxacin antibiotic (5–10 mg/kg subcutaneously) was given immediately before surgery began.

A 4-cm longitudinal incision was made with a No. 10 scalpel. Paraspinal muscles were released subperiosteally. Lateral gutters were made at the facet joint border to open the lamina with a microtome. The lamina was then opened like a door. A spacer was placed to support the elevated lamina. Soft tissue and skin were closed. Rabbits were returned to the cage and observed. Analgesics were given immediately after surgery and 12 h postoperatively.

Euthanasia was performed 12 weeks postoperatively by intravenous injection of phenobarbital (10 mg/kg) after blood samples from the ear were taken to evaluate systemic tissue reactions. The previous surgical scar incision was opened. Vertebrae were taken from several levels and removed from the body of a rabbit, then divided in two at the middle of the scaffold. Samples were prepared to be sent to the anatomy pathology lab and to be freeze-dried before being sent off for SEM examination.

### 2.8. Evaluation

Evaluation was carried out by assessing the immune response from both cellular aspects (histopathological examination) and serological reactions (erythrocyte sedimentation rate (ESR)). Architectural assessments of pre- and post-implantation scaffolds were done by scanning electron microscopy (SEM).

### 2.9. Histopathology

Samples were placed in 10% formalin to make histopathological slide preparations using hematoxylin–eosin staining. Preparation of the tissue for removal started by washing the slide in running water, preceded by decalcification for 24 h using a Shandon TBD-1 Rapid Decalcifier (Epredia, Runcorn, UK) solution and continued washing with running water. Then, the tissue was cut, and the scaffold was attached macroscopically and placed on a cassette. The tapes were then soaked in running water for 60 min and then, in succession, in 80% alcohol for 24 h, 90% alcohol for 30–60 min, 100% ethanol for 30 min as many as six times, and then Xylol solution for 30 min.

Macroscopic tissue was then put into 600 °C liquid paraffin for 1 h. The cassette block was then cooled to room temperature for 15 min until it hardened and cut with a 3-μm microtome, 500–1000 um per piece. After settling for one day, HE color was administered.

Cell counting was conducted using ImageJ cell counting software. From the number of cells obtained, the degree of inflammation can be determined. We use the lowest magnification to scan the area with the inflammatory cells. Determination of the degree of inflammation follows the following assessment:
(1)1st degree: no inflammatory cells per large visual field;(2)2nd degree: there are <25 inflammatory cells per large visual field;(3)3rd degree: there are 25–125 inflammatory cells per large visual field;(4)4th degree: there are >125 inflammatory cells per large visual field;(5)5th degree: there is tissue necrosis.


### 2.10. Scanning Electrone Microscope

Prior to SEM examination, samples were prepared by the freeze-drying method of the National Nuclear Energy Agency (BATAN) Isotope and Radiation Technology Application Testing Laboratory (Yogyakarta, Indonesia). After drying, a coating process was carried out with a conductive material because the sample is a nonconductive material. In this research, gold was used as a coating.

### 2.11. Physicochemical Properties

#### 2.11.1. TGA and DSC Analysis

Thermogravimetric analysis (TGA) was used to describe the thermal stability of the realized scaffolds. A simultaneous thermal analyzer (Labsys Evo 1600 Setaram Instrumentation, Caluire, France) was utilized. The sample was heated from 25 to 600 °C at a heating rate of 20 °C/min. The mass change data were analyzed to estimate the temperature of the onset of degradation (Tonset) of the composite.

The Differential Scanning Calorimeter mode was conducted using the same equipment to observe the thermal transition of the compound. The sample was heated from 0 to 200 °C at a heating rate of 10 °C/min. The measurement resulted in the glass transition temperature (Tg), cold crystallization temperature (Tcc) and melt temperature peak (Tm).

#### 2.11.2. Fourier Transform Infrared Red Spectroscopy (FTIR)

The FTIR spectra were recorded on a Fourier Transform Infra-Red Nicolet™ iS50 Spectrometer (Thermo Fisher Scientific, Waltham, MA, USA) in the spectral range of 15–27,000 cm^−1^. The result of transmittance was compared with literature values to determine its functional groups.

#### 2.11.3. X-ray Diffraction (XRD) Analysis

X-ray diffraction was utilized to observe the microcrystalline structure. A XRD Panalytical Empyrean (Malvern Panalytical, Malvern, UK) at 40 kV and 30 mA with Cu anode radiation at wavelength of 1.5406 Å was set at room temperature. The result analyzed using HighScore Plus Version 3.0 Standpoint with 2θ between 10° and 80° with a 0.0263°/step scanning rate.

## 3. Results

At the two-week follow-up, three rabbits were dead and replaced by other rabbits. Two of the dead rabbits experienced post-operative paralysis. No rabbits developed infection.

### 3.1. Systemic Tissue Reaction Evaluation

ESR was evaluated before implantation to determine the baseline value and after implantation to evaluate the tissue reaction to the scaffold. The median baseline value was 2 (range: 1–3) and after implantation the value was 1 (1–4). There was no significant difference in the ESR values before and after implantation (*p* = 0.442). Intergroup analysis also showed no significant difference between groups (*p* = 0.845) (Figure 2).

### 3.2. Local Tissue Reaction Evaluation

Histopathological examination was performed by H&E staining. The tissue reaction was evident from the increase in the number of inflammatory cells, which consisted of adaptive immunologic cells (plasma cells and lymphocytes). After identification, those cells were counted in ImageJ software. Generally, no inflammatory reaction (grade 1) was found in the majority of samples (47%). No sample (0%) presented with a necrotic area (grade 5) (Figure 3).

The control group had the lowest inflammatory grade (grade 1). Scaffold groups had a variable inflammatory reaction with no significant difference between them and no significant difference compared to the control group (*p* = 0.505). A correlation study using an exact Fisher test was performed to evaluate the correlation of the intervention group and histologic inflammatory grading. We found no correlation between the intervention groups and histologic grading (*p* = 0.505).

### 3.3. Evaluation of Physicochemical Properties

The Differential Scanning Calorimeter mode was conducted using the same equipment to observe the thermal transition of the compound. The sample was heated from 0 to 200 °C at a heating rate of 10 °C/min. The measurement resulted in the glass transition temperature (Tg), cold crystallization temperature (Tcc), and melt temperature peak (Tm). Figure 4a indicated Tg, Tcc and Tm were 51 °C, 97 °C and 160 °C, respectively.

As shown in Figure 4b, the thermaogravimetry analysis indicates that there is no significant decomposition from room temperature to 300 °C, suggesting that the PLA component hindered degradation of the composite. The PLA filament as the predominant component has been demonstrated to have a thermal decomposition temperature ranging 300–370 °C, which is also within the same range of alginate decomposition temperature.

Exothermic peak at 100 °C was observed due to the evaporation of the remaining solution of the injectable HA/alginate, followed by an endothermic peak caused by tightly covalent binding in alginate around 160–170 °C. Within the alginate molecular network, hydrogen bonds potentially formed in the presence of carbonyl and hydroxyl groups. Mixing of HA to the alginate as the filler could also be a factor to stabilize the physical behavior of alginate due to interactions between calcium from hydroxyapatite to sodium alginate.

The interaction between the function group of polymers was observed as in infrared spectra in Figure 5. The infrared (IR) spectra at 2995 and 2945 cm^−1^, which indicated the asymmetric and symmetric C–H stretching region, were identified. The strong absorption band at 1747 cm^−1^ was characterized for the C=O stretching peak of ester group. The characteristics of stretching peaks at 1180 cm^−1^ indicated the presence of C–O. The strong band at 1080 of C–O–C stretching is also indicated in our observation.

The diffraction patterns indicated the crystallinity phase of PLAas, as shown in Figure 5a. The diffraction angle, 2θ, was presented from 5° to 80°. The XRD pattern for the PLA did not show any characteristic peak, which indicates that the structure of PLA is amorphous. The XRD pattern of PLA exhibited a low diffraction peak at around 10–20°, alginate—HA has been observed in our previous study (Lestari et al., 2017) and demonstrated an amorphous structure.

### 3.4. Evaluation of Scaffold Microstructure

After implantation, it was expected that the scaffold would be biocompatible and nontoxic to body tissues, as indicated by the presence of tissue integration with the scaffold. A three-dimensional PLA scaffold was designed with 41% porosity, and the biocompatibility profile was confirmed by an in vitro study. After 12 weeks of implantation, the PLA scaffold was still identifiable macroscopically. This made it easier for identification in further examinations. Scaffolds were then processed for further evaluation with a scanning electron microscope.

We evaluated the porosity of the scaffolds and compared it with the baseline porosity using ImageJ [29]. A reduction in porosity demonstrated tissue integration with the porous scaffold (Figure 6). The porosity distribution was summarized in Table 1. Most of the samples (12 out of 15) showed reduced scaffold porosity, which indicated tissue integration with the scaffold. There was no significant mean difference between groups (*p* = 0.361). There was no significant difference between the intervention and control group porosity (*p* = 0.101) or between the scaffold-only group (group 2) and the augmentation group (groups 3–5) (*p* = 0.249). There was no correlation between porosity and histological grading (*p* = 0.093).

## 4. Discussion

Cervical myelopathy is a chronic degenerative disease that causes spinal cord compression at the cervical level. Several decompression methods have been proposed. Laminoplasty is one decompression technique using a posterior approach that is performed by reconstructing part of the lamina and creating a hinge to widen the canal. Several techniques have been developed to maintain the open hinge, such as using implant fixation or filling the gap with a spacer. The use of a plate is associated with hinge fusion failure due to the absence of ventral cortical bone and low adaptability to the bone tissues. Compared to autografts, spacers can shorten the operating time, reduce morbidity, and shorten the post-operation neck immobilization period. The use of autografts originating from the spinous process, iliac crest, or ribs is associated with higher levels of donor site morbidity, risk of infection, bleeding, graft kick-out and more postoperative pain. Alternatively, recent studies have focused on the use of biomaterials to prevent these graft-associated complications [1,2,3].

A biomaterial is a material that can safely interact with living tissue. It is designed to meet medical needs, either diagnostic or therapeutic. The biomaterial’s capacity to be safely introduced to a living tissue of a certain species without causing a destructive or toxic reaction is the meaning of biocompatibility. Biocompatibility was determined in a tissue reaction by the presence of inflammatory reaction and tissue integration [16,17,30,31,32,33].

Table 2 shown a comparative study of our fabricated scaffold with similar works. It can be concluded that the line widths were in a similar range since the fabrication method is similar. We found a similar porosity in our scaffold as with Naghieh et al. Our previous result demonstrates that the porosity was initially at 60% and 40% when filled with the HA-alginate. However, this work also indicates that the mechanical properties heavily affected the PLA material and structure. Therefore, it was found that our elastic modulus resulted in a 90 MPa. This result is not that different from the PLA printed by Naghieh et al. with porosity of 40%. The physico chemical properties were in accordance with the Nascimento group et al. for the PLA part. The transition temperature (Tg), cold crystallization temperature (Tcc) and melt temperature peak (Tm) of PLA also agreed with the result of Baptista, Vanaei and Nascimento. The interaction between PLA-HA was also demonstrated by the Arastouci and exhibited an amorphous behavior.

In this study, we reported the clinical results of a biocompatibility assessment using ESR as systemic parameters of tissue reaction, the presence of inflammatory cells in HE-stained samples, the evaluation of tissue integration by SEM and a porosity evaluation. There has been a lack of studies that analyze and study stem cells in a cell-based construct, such as that used in the laminoplasty procedure. However, those studies reported that the use of cell-based scaffolds involves a higher level of mineralization and ossification compared to cell-free approaches; thus, this technique has become clinically advantageous for stimulating bone regeneration and healing processes. A recent study by Reichert et al. demonstrated the satisfactory outcome using a stem cell-rich scaffold in combination with scaffolds for managing bone defects [20,31,33,40].

ESR is a method to determine the sedimentation of blood in a test tube, measured in mm/h. The increase in ESR indicates a systemic inflammatory reaction and is maintained for a period of time in the blood before the value drops to the baseline. A high ESR value or tendency to increase compared to the baseline can be a valuable parameter to predict tissue rejection. In our study, ESR values before and after implantation exhibited no significant difference (*p* = 0.442). This indicates that there was no or minimal systemic tissue reaction, which did not differ significantly from the baseline [41,42,43].

In an acute inflammatory reaction, inflammatory cells such as monocytes, macrophages, and polymorphonuclear (PMN) leukocytes are recruited and accumulate on the material and at the interface between the material and the host tissue [31,44]. The inflammatory reaction associated with the material application can last from a few days to a couple of weeks depending on the material type, site of application, and soft tissue damage associated with the implantation procedure [3,45].

In this study, we categorized the inflammatory reaction degree into five grades based on the chronic inflammatory cell count [3,46]. This semiquantitative scoring system was set up to evaluate the degree of cell death in response to a biomaterial application [3,46]. In our study, we found that there was no significant difference in tissue reaction grading among scaffolds (*p* = 0.505). Autografts were expected to have the best biocompatibility without inflammatory reaction. This was corroborated by our finding that the control group had the lowest inflammatory grade (grade 1). Scaffold groups had a variable inflammatory reaction, with no significant difference among them and no significant difference compared to the control group (*p* = 0.505). These important results demonstrated that the scaffold application is as biocompatible as autografts and can be used in the next level of a study with human subjects. We also found that there was no correlation between the intervention group and histologic grading (*p* = 0.505). This suggests that the augmentation of the scaffold did not correlate with the tissue reaction.

This result demonstrated that the laminoplasty spacer using a biomaterial has the same systemic and local tissue reaction profile as the autograft spacer, with no risk of graft failure or donor-site morbidity. Therefore, the use of 3D biomaterial combined with stem cell or HA can be applied for laminoplasty spacer in further study in humans.

According to the diamond concept model, MSCs act as progenitor cells that are expected to form new bone tissue and express the growth factors needed to trigger the surrounding host cell to grow in osteogenic lineage. Thus, the presence of both osteogenic and osteo-inductive properties in the scaffold will contribute to both fusion and bone union processes. The scaffold itself provides 3D cues for the formation and development of tissue implantation [21,22,47,48].

Tissue integration with the scaffold can be evaluated using a scanning electron microscope (SEM) and porosity can be evaluated using ImageJ [18,29]. There was a reduced porosity in most of the scaffolds (12 of 15) compared to the baseline, although the difference was not significant, which might have been caused by the small sample in this study. This result demonstrated that polymer-based scaffolds were as good as autografts in terms of tissue integration and might be even better, as shown by the lower median value in the scaffold group. We also compared group 2 (PLA-only scaffold) and groups 3–5, which consisted of an augmented scaffold (40% vs. 15%). This might indicate that augmentation using MScs and osteoconductive substances can increase tissue integration.

## 5. Conclusions

PLA-based scaffolds, with or without augmentation with HA/alginate and MSCs, obtained similar results in terms of tissue reaction and tissue integration with the autograft. The systemic tissue reaction, evaluated by the ESR values before and after implantation, demonstrated no significant difference in all groups. This could indicate that there was no or only minimal systemic tissue reaction in all groups, which did not differ significantly from the baseline. In evaluating the local tissue reaction, the autografts were expected to have the best biocompatibility without significant rejection or inflammatory reaction. This was also proven in this study, in which the control group had the lowest inflammatory grade (grade 1). Although scaffold groups had variable inflammatory reactions, there was no apparent difference compared to the control group. This result could demonstrate that the scaffold application is as biocompatible as autografts and can be used in the next level of a study with human subjects. In order to prove their advantages, we encourage further studies such as clinical trials be performed so that this biomaterial can be implemented as a laminoplasty spacer.

## Figures and Tables

**Figure 1 polymers-14-03292-f001:**
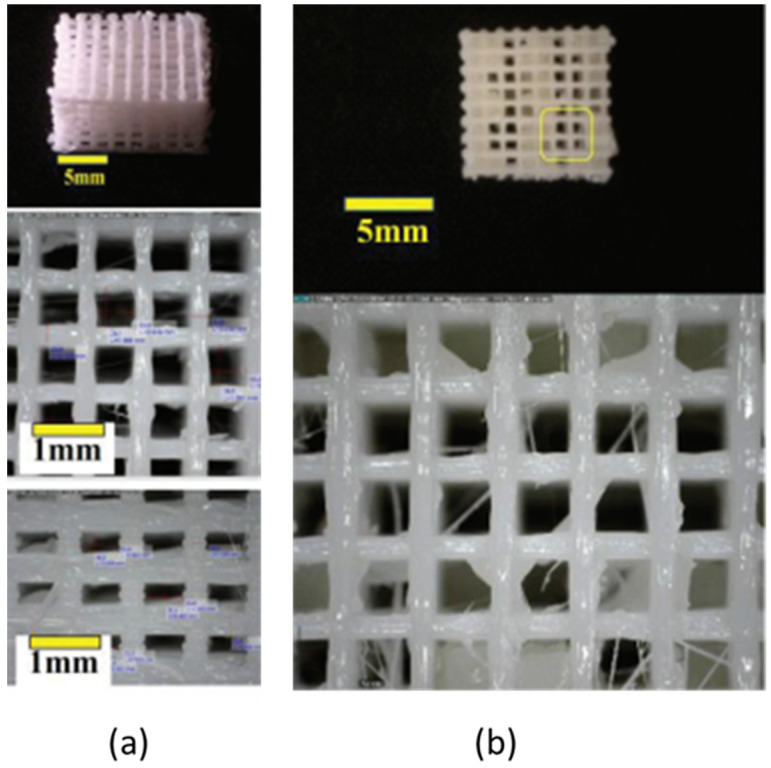
(**a**) Six-layer 3D PLA scaffold; (**b**) post hydroxyapatite (HA)/alginate composite injection to six-layer 3D PLA scaffold.

**Figure 2 polymers-14-03292-f002:**
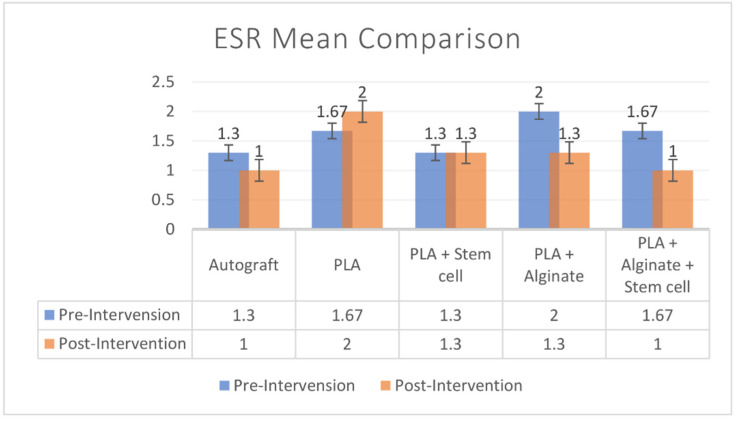
Pre-operative and post-operative ESR comparison among groups.

**Figure 3 polymers-14-03292-f003:**
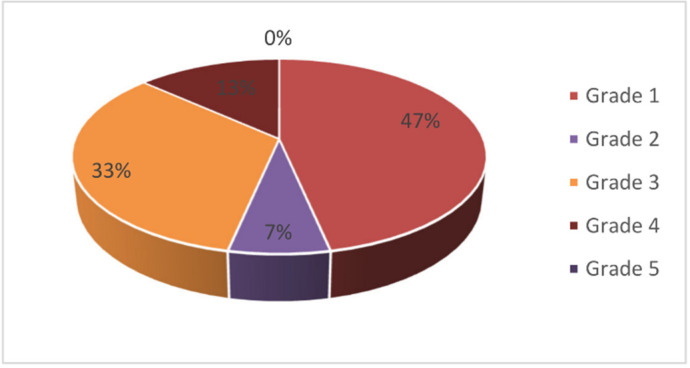
Distribution of histological tissue reaction grading: 47% of samples were at grade 1; 7% at grade 2; 33% at grade 3, 13% at grade 4, and 0% at grade 5.

**Figure 4 polymers-14-03292-f004:**
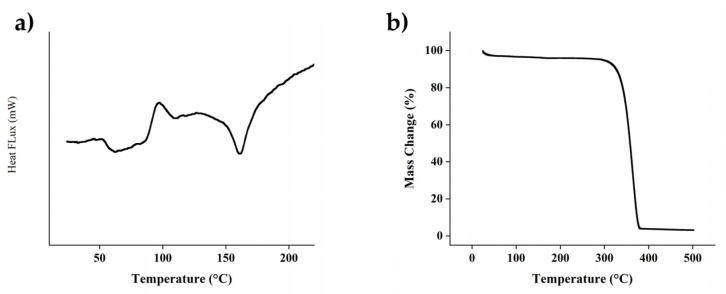
Result of thermal characterization using (**a**) Differential Scanning Calorimetry and Thermal Gravimetric Analysis for PLA. (**b**) Thermaogravimetry Analysis for PLA.

**Figure 5 polymers-14-03292-f005:**
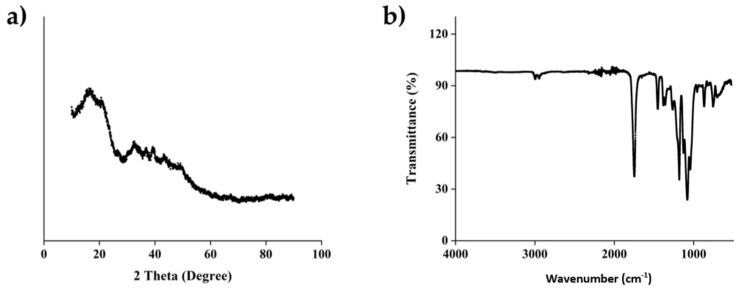
(**a**) The crystalline phase and molecular interaction and (**b**) Fourier Transform infrared wave spectrum of PLA.

**Figure 6 polymers-14-03292-f006:**
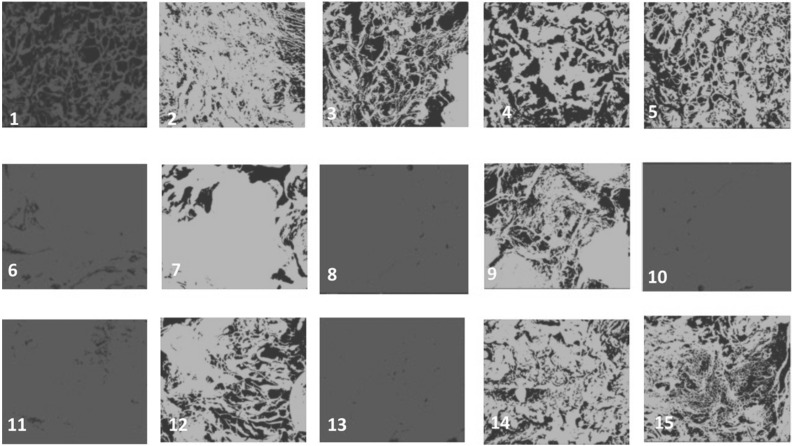
Microstructure and porosity evaluation with scanning electron microscope and ImageJ tools 12 weeks after implantation showed tissue integration with scaffolds: (**1**–**3**) Group 1 (autograft), (**4**–**6**) Group 2 (PLA), (**7**–**9**) Group 3 (PLA + HA/alginate), (**10**–**12**) Group 4 (PLA + MSc) and (**13**–**15**) Group 5 (PLA + HA/alginate + MSc).

**Table 1 polymers-14-03292-t001:** Porosity distribution among groups.

Variable Porosity (%)	Value (Mean ± SD)
Group 1	39.67 ± 18.34
Group 2	28.67 ± 25.01
Group 3	22.67 ± 20.52
Group 4	28 ± 18.34
Group 5	15 ± 15

**Table 2 polymers-14-03292-t002:** Comparison of a similar study.

Type of Scaffolds	Line Width	Porosity (%)	Elastic Modulus	Ref.
3D-printed PLA-HA-alginate	400 µm	40–60%	90 MPa	This work
3D-printed PLA-gelatin-forsterite	500 µm	39	112 MPa	Naghieh 2017 [34]
FDM PLA	NA	NA	246 MPa	Nascimento 2021 [35]
3D-printed PCL	NA	NA	30 MPa	Schipani 2019 [36]
3D-printed PLA	100–400 µm	70–30	50–600 MPa	Baptista 2021 [37]
PLA-HA	150 µm	NA	24 MPa	Arastouei 2020 [38]
2D-printed PLA	600 µm	NA	500 ± 10 MPa.	Vanaei 2020 [39]

## Data Availability

The data presented in this study are openly available in Dryad at https://doi.org/10.5061/dryad.r7sqv9s91.

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
