# Peer review of "The Use of 3D Polylactic Acid Scaffolds with Hydroxyapatite/Alginate Composite Injection and Mesenchymal Stem Cells as Laminoplasty Spacers in Rabbits"

_polymers, 2022, doi:10.3390/polym14163292_

Round 1

Reviewer 1 Report

In this article, the authors report the synthesis and application of a polylactic acid (PLA) composite scaffold as laminoplasty spacer for filling bone gaps. The results are of possible interest to the fields of implant materials and additive manufacturing.

My comments are provided below, with major recommendations for the introduction and discussion/conclusion sections.

Introduction section

I found this section technically sound. One key missing aspect is the definition of an ideal composite material for implant purposes. While this study focusses on biocompatibility and histopathology, other factors also determine the success of implant materials in general.  Especially for bone-mimetic composites, certain key requirement must be met regarding microstructure, nanostructure, biomineralization, re-adsorption and re-modelling. I recommend the authors introduce this additional motivation of their work. This can be supported by key review articles such as (i) "The components of bone and what they can teach us about regeneration." Materials 11.1 (2017): 14., (ii) What should be the characteristics of the ideal bone graft substitute?. Injury, 42, pp.S1-S2. (iii) 2016. Morphology control and molecular templates in biomineralization. In Biomineralization and Biomaterials (pp. 51-93). Woodhead Publishing. etc

Methods section

  1. Details of ‘open-door laminoplasty’ procedure are missing. (line 87)
  2. After the injection, the porosity of the injected scaffolds was measured using the same method. (line 137) …. The method principle to measure porosity is not described. I presume it is done using an imaging software, additional details must be added.
  3. I miss mention of culture media and their compositions in cell biology experiments
  4. Section 2.9: define ‘large field of view’. What was the area analyzed?

Results section

  1. Figure 2: Good that the authors applied statistical method. However, the figure 2 lacks error bars.
  2. Line 337 : Therefore, for further study in humans it is necessary to make this biomaterial applicable to laminoplasty spacers…which material is referred to here? PLA? + Ha? + ALg?
  3.  Readability must be improved in general.

Discussion and Conclusion sections

A main criticism is the lack of discussion. How does the described process and material properties compare with previously studies ones?

Based on a brief literature survey, I could find recent studies on PLA composites for biomaterial application. I strongly suggest that the authors compare their findings and refer  to literature reports on biocompatibility, mechanics and other properties. For examples, see (i)  Preparation and characterization of electrospun alginate/PLA nanofibers as tissue engineering material by emulsion eletrospinning. Journal of the mechanical behavior of biomedical materials. 2017 Jan 1;65:428-38., (ii) 3D-bioprinting of polylactic acid (PLA) nanofiber–alginate hydrogel bioink containing human adipose-derived stem cells. ACS biomaterials science & engineering, 2(10), pp.1732-1742. etc

Also, what is your future outlook for these composite materials? (in light of your observations)

Author Response

Response to Reviewer’s Comments

Point 1

I found this section technically sound. One key missing aspect is the definition of an ideal composite material for implant purposes. While this study focusses on biocompatibility and histopathology, other factors also determine the success of implant materials in general.  Especially for bone-mimetic composites, certain key requirement must be met regarding microstructure, nanostructure, biomineralization, re-adsorption and re-modelling. I recommend the authors introduce this additional motivation of their work. This can be supported by key review articles such as (i) "The components of bone and what they can teach us about regeneration." Materials 11.1 (2017): 14., (ii) What should be the characteristics of the ideal bone graft substitute?. Injury, 42, pp.S1-S2. (iii) 2016. Morphology control and molecular templates in biomineralization. In Biomineralization and Biomaterials (pp. 51-93). Woodhead Publishing. Etc

Response 1

Thank you for the comments. We already added more urgency on how biomaterials can fill the gap between current option of autograft and allograft.

Point 2

Details of ‘open-door laminoplasty’ procedure are missing. (line 87)

Response 2

The detail of the ODL was stated in the Section 2.7. We decided to erase the sentences previously on line 87 and to explain the laminoplasty procedure on section 2.7

Point 3

After the injection, the porosity of the injected scaffolds was measured using the same method. (line 137) …. The method principle to measure porosity is not described. I presume it is done using an imaging software, additional details must be added.

Response 3

Thank you for the review. We put additional information on the method to analyze the porosity, as follows.

After the injection, the porosity of the injected scaffolds was measured using the same method by using ImageJ (version 1.47 for Windows, 64 bit, free software, National In-stitutes of Health, Bethesda, MD, USA). After thresholding and binarization was done, the porosity was determined using image volume method.

Point 4

I miss mention of culture media and their compositions in cell biology experiments

Response 4

In this experiment alpha MEM and Dulbecco’s modified Eagle’s medium (DMEM,[GIBCO 31600-034]) were used as the medium.

Point 5

Section 2.9: define ‘large field of view’. What was the area analysed?

Response 5

We corrected the wording to large visual field. Meaning using the lowest magnification to search for the inflamed part of the tissue.

Point 6

Figure 2: Good that the authors applied statistical method. However, the figure 2 lacks error bars.

Response 6

Thank you for the comment. We already added the error bars to the figure 2 on the revision.

Point 7

Line 337 : Therefore, for further study in humans it is necessary to make this biomaterial applicable to laminoplasty spacers…which material is referred to here? PLA? + Ha? + ALg?

Response 7

Thank you for the comments. What we wanted to state was how the 3D biomaterial spacer with or without additional substance like HA or MSC had the same local or systemic reaction compared to the gold standard that is the autograft.

Point 8

Readability must be improved in general.

Response 8

Thank you for the review. We try to improve our readability by using the language editor provided by this journal. We also revised several sentences and add several points in our revised article. Hopefully it will be suffice.

Point 9

A main criticism is the lack of discussion. How does the described process and material properties compare with previously studies ones?

Based on a brief literature survey, I could find recent studies on PLA composites for biomaterial application. I strongly suggest that the authors compare their findings and refer  to literature reports on biocompatibility, mechanics and other properties.

For examples, see (i)  Preparation and characterization of electrospun alginate/PLA nanofibers as tissue engineering material by emulsion eletrospinning. Journal of the mechanical behavior of biomedical materials. 2017 Jan 1;65:428-38.,

 (ii) 3D-bioprinting of polylactic acid (PLA) nanofiber–alginate hydrogel bioink containing human adipose-derived stem cells. ACS biomaterials science & engineering, 2(10), pp.1732-1742. etc

Response 9

We thank the reviewer for giving us opportunity to clarify on how our study linked to the similar studies. Therefore, we have updated the manuscript to make the innovations clearer.

Our previous work (Syuhada et al 2018) showed the preparation and mechanical characteristic of realized scaffolds. The elastic modulus was achieved around 90 MPa with porosity around 40%. The composition was polylactic acid filled with HA.  

The toxicity was validated with other study (Rahyussalim 2017) showing that the scaffold PLA-HA were proven to be non-cytotoxic. The HA filler itself was also has been characterized (Ruri et al 2017 and Albab 2018 et al). Based on Naghieh 2017, their latest research was founded that the EM at around 110 MPa which similar to our result (Naghieh 2017 https://doi.org/10.1016/j.matdes.2017.07.051). Their porosity was also similar (40%).

We did also add a comparative study to put our work with other similar groups. Our physicochemical properties agreed with other especially on the PLA and HA part. The uniqueness that we brought here is the HA-alginate insertion 

Table 2 Comparation of similar study

Type of scaffolds

Line width

Porosity (%)

Elastic Modulus

Physico-chemical properties

Ref.

3D printed PLA-HA-alginate

400um

40-60%

90 MPa

FTIR, XRD TGA DCS of PLA

HA-alginate

This work

Ruri 2017

3D printed PLA-gelatin-forsterite

500um

39

112 MPa

NA

Naghieh 2017

FDM PLA

NA

NA

246 MPa

FTIR, XRD TGA DCS matchd

Nascimento 2021

3D printed PCL

NA

NA

30 MPa

Scipani 2019

3D printed PLA

100-400um

70-30

50-600 MPa

DCS matched

Baptista 2021

PLA-HA

150um

NA

24 MPa

XRD matched

Arastouci 2020

2D printed PLA

600um

NA

500±10 MPa.

DSC matched

Vanaei 2020

Supporting reference:

Investigating the weight ratio variation of alginate-hydroxyapatite composites for

vertebroplasty method bone filler material, Gusti Ruri Lestari, Akhmad Herman Yuwono, Nofrijon Sofyan, and Ghiska Ramahdita, Citation: AIP Conference Proceedings 1817, 020006 (2017); doi: 10.1063/1.4976758

Combinational processing of 3D printing and electrospinning of hierarchical poly(lactic acid)/gelatin-forsterite scaffolds as a biocomposite: Mechanical and biological assessment

Saman Naghieh, Ehsan Foroozmehr, Mohsen Badrossamay, Mahshid Kharaziha

Toward the understanding of temperature effect on bonding strength, dimensions and geometry of 3Dprinted parts, H. R. Vanaei, and A. Tcharkhtchi, K. Raissi, M. Deligant, M. Shirinbayan,  J. Fitoussi, S. Khelladi

Improving the Properties of the Porous Polylactic Acid Scaffold by Akermanite Nanoparticles for Bone Tissue Engineering, Masoud Arastouei, Mohammad Khodaei,

Sayed Mohammad Atyabi, Milad Jafari Nodoushan

Morphological and mechanical characterization of 3D printed PLA scaffolds with controlled porosity for trabecular bone tissue replacement, R. Baptista and , M. Guedes

Integrating finite element modelling and 3D printing to engineer biomimetic

polymeric scaffolds for tissue engineering, Rossana Schipani, David R. Nolan, Caitr#ona Lally & Daniel J., Kelly (2020)  Connective Tissue Research, 61:2, 174-189, DOI:

10.1080/03008207.2019.1656720

The influence of phosphorylation and freezing temperature on the mechanical properties of hydroxyapatite/chitosan composite as bone scaffold biomaterial Muh Fadhil Albab, Nicholas Giovani, Akhmad Herman Yuwono, Nofrijon Sofyan, Ghiska Ramahdita, and Yudan Whulanza

Multi-material poly(lactic acid) scaffold fabricated via fused deposition modeling and

direct hydroxyapatite injection as spacers in laminoplasty Ghifari Syuhada, Ghiska Ramahdita, A. J. Rahyussalim, and Yudan Whulanza

Influence of the printing parameters on the properties of Poly(lactic acid) scaffolds

obtained by fused deposition modeling 3D printing, Abraão CD Nascimento Jr, Raquel CDAG Mota, Livia RD Menezes, and Emerson OD Silva

Point 10

Also, what is your future outlook for these composite materials? (in light of your observations)

Response 10

We brought an idea that our scaffolds able to deliver a customised structural and suffice mechanical characteristic of seeded cells that delivered with the PLA material. The seeded cells would be accommodated with the alginate to support the viability of cells. Furthermore, the HA part shall deliver the osteocondtivity altogether with improving the mechanical characteristic as designed.

Reviewer 2 Report

In this manuscript, the authors evaluated the in vivo biocompatibility and tissue–scaffold integration of a polylactic acid (PLA) scaffold with the addition of alginate/hydroxyapatite (HA) and mesenchymal stem cell (MSc) injections. However, very limited data are presented. In particular, the scientific contribution of the manuscript is NOT significant, and the novelty of the work is under question. I must reject it.

Author Response

Point 1

In this manuscript, the authors evaluated the in vivo biocompatibility and tissue–scaffold integration of a polylactic acid (PLA) scaffold with the addition of alginate/hydroxyapatite (HA) and mesenchymal stem cell (MSc) injections. However, very limited data are presented. In particular, the scientific contribution of the manuscript is NOT significant, and the novelty of the work is under question. I must reject it.

Response 1

Thank you for the comment. 

Our research novelty is on how our 3D PLA scaffold could be used with or without MSC, HA/Alginate, or both as the laminoplasty spacer in Animal models. In the future, we would like to continue our study with human subjects. 

We revised the manuscript according to the reviewer's comments. Hopefully, the reviewer would consider evaluating the revised manuscript.